# Nearly 20 Years of Genetic Diversity and Evolution of Porcine Circovirus-like Virus P1 from China

**DOI:** 10.3390/v14040696

**Published:** 2022-03-28

**Authors:** Libin Wen, Lihong Yin, Jiaping Zhu, Heran Li, Fengxi Zhang, Qun Hu, Qi Xiao, Jianping Xie, Kongwang He

**Affiliations:** 1Institute of Veterinary Medicine, Jiangsu Academy of Agricultural Sciences, Nanjing 210014, China; yinlihong2021@163.com (L.Y.); zhujp0206@yeah.net (J.Z.); 2020107095@stu.njau.edu.cn (H.L.); 17767735839@163.com (F.Z.); 13834986741@163.com (Q.H.); xiaoqi2122@163.com (Q.X.); xiejp1969@163.com (J.X.); 2Key Laboratory of Veterinary Biological Engineering and Technology, Ministry of Agriculture, Nanjing 210014, China; 3Jiangsu Co-Innovation Center for Prevention and Control of Important Animal Infections Diseases and Zoonoses, Yangzhou University, Yangzhou 225009, China; 4Jiangsu Key Laboratory for Food Quality and Safety—State Key Laboratory Cultivation Base of Ministry of Science and Technology, Nanjing 210014, China

**Keywords:** porcine circovirus-like virus P1, genetic diversity, evolution, recombination, China

## Abstract

Porcine circovirus-like virus P1 can infect many kinds of animals and mainly causes postweaning multisystemic wasting syndrome. In China, the genetic diversity, variation, and evolutionary processes of this virus have not been described yet. To improve our knowledge of its genetic diversity, evolution, and gene flow, we performed a bioinformatics analysis using the available nucleotide sequences of the P1 virus; among them, 12 nucleotide sequences were from ten pig farms in Jiangsu Province in this epidemiological survey, and 84 sequences were downloaded from GenBank. The P1 sequences showed a rich composition of AT nucleotides. Analyses of the complete genomic sequences were polymorphic and revealed high haplotype (gene) diversity and nucleotide diversity. A phylogenetic analysis based on the NJ method showed that all P1 virus sequences formed two distinct groups: A and B. High genetic differentiation was observed between strains from groups A and B. The codon usage pattern of P1 was affected by dinucleotide compositions. Dinucleotide UU/CC was overrepresented, and dinucleotide CG was underrepresented. The mean evolutionary rate of the P1 virus was estimated to be 3.64 × 10^−4^ nucleotide substitutions per site per year (subs/site/year). The neutrality tests showed negative values. The purifying selection and recombination events may play a major driving role in generating the genetic diversity of the P1 population. The information from this research may be helpful to obtain new insights into the evolution of P1.

## 1. Introduction

Porcine circovirus-like virus P1, belonging to the family *Circoviridae,* may be another causative agent of PCV2-associated disease, which is characterized by progressive weight loss, wasting, respiratory signs, enteritis, tremors, and reproductive failure, among other features [1,2,3,4]. Since its discovery in 2005, the P1 virus has been common in pig farms in China [5,6,7,8,9]. The P1 virus was found in various animals, such as pigs, dogs, cats, rabbits, goats, cattle, and yaks [2,10,11].

The P1 virus is a closed, circular single-stranded DNA virus with a genome size of 648 nt, making it the animal virus with the smallest genome by far. The other PCV genomes are about 1770 to 2000 nt. Compared with other PCVs, the P1 virus shows the highest genomic identity to PCV2 ORF2 (96.0%–97.0%). The genome of the P1 virus contains eight potential open reading frames (ORFs), of which three ORFs (ORF1-3) have been confirmed [12,13]. The largest ORF1 encodes a Cap protein containing 114 amino acids and is the main immunogen, which is also suitable for P1 virus mutation analysis. The ORF2 and ORF3 are mainly associated with biological processes, such as the positive regulation of leukocyte chemotaxis, cell proliferation, cell migration, the defense response to the virus, and the regulation of cell growth [14]. With the increasing number of reported P1 strains in recent years, some novel strains with different characteristics have been detected one after another. For example, a P1 virus strain containing a 122-amino acid capsid protein, 8 amino acids longer than the capsid protein of the original strain, was detected in 2015 [15]. In 2017, a strain with a genome length of 647 nucleotides appeared, and its capsid protein contains 90 amino acids [16]. Recently, P1 strains with a genome length of 649 nucleotides and a capsid protein containing 163 amino acids were reported [17].

In the past few years, research has mainly focused on P1 epidemiology. So far, 84 complete P1 genomes have been submitted to the GenBank database, but there is no report on the genetic features and evolutionary processes of this emerging circovirus in China. In this study, we investigated the prevalence of P1 using 240 clinical pig samples collected from ten farms in Jiangsu Province, China, between 2020 and 2021, for routine diagnosis. In addition, the nucleotide sequences of all P1 virus strains available in GenBank were used for analysis. We aimed to elucidate the different P1 phylogroups, their genetic traits, and their evolution in China.

## 2. Materials and Methods

### 2.1. Serum Sampzles and Extraction of Genomic DNA

A total of 240 serum samples were collected from clinically healthy pigs from ten farms in Jiangsu Province between January 2020 and July 2021. Genomic DNA was extracted using a commercial kit (TIANDZ, China) according to the manufacturer’s protocol. All of the experimental protocols using animals were conducted according to the Jiangsu Province Animal Regulation Guidelines (Government Decree No 45) and were approved by the Jiangsu Academy of Agricultural Sciences Experimental Animal Ethics Committee (No. NKYVET 2015-0126).

### 2.2. Amplification and Sequencing Analysis of Porcine Circovirus-Like Virus P1

The full-length genome of the P1 virus was amplified by a PCR method, as described previously [11]. The high-fidelity enzyme (Takara, Dalian, China) was used in all PCR amplifications to minimize the nucleotide misincorporation. PCR products were assessed via electrophoresis in an 1.5% agarose gel and visualized with ultraviolet light.

Each PCR product was purified from the gel, cloned into the pMD18-T vector (Takara, Dalian, China), and the ligation products were transformed into *Escherichia coli* DH5α competent cells. The positive recombinant clones were selected via colony PCR and sequenced from both directions using standard Sanger sequencing. Plasmids from at least three independent clones from each transformation mixture were sequenced to verify the sequence accuracy.

### 2.3. Data Analysis

All 12 P1 virus sequences obtained in this study and 84 P1 nucleotide sequences were retrieved from the GenBank database that had been submitted up to December 2021 (Appendix A), and they were analyzed using MegAlign in the DNASTAR package (DNASTAR Inc., Madison, WI, USA, Version 15.0.0). After removing the redundant sequences, the remaining 83 P1 nucleotide sequences were used for subsequent analysis.

A phylogenetic tree was created using MEGA7 (Version 5.03) by using the NJ method, and bootstrap values were computed with 1000 bootstrap replicates.

DNA sequence polymorphisms, including the value of singleton variable sites, parsimony informative sites, the total number of mutations, the average number pairwise nucleotide difference (K), haplotype diversity (Hd), nucleotide diversity (π), and the rate of synonymous (dS) and non-synonymous (dN) substitutions for each population, were calculated using DnaSP version 5.10.01. The π value was also measured on a sliding window plot of 10 bases with a step size of 5 bp to estimate the stepwise diversity across the sequences. The DNA divergence between populations was also measured on a sliding window plot of 100 bases with a step size of 25 bp. Genetic differentiation on the full length of the Cap-encoding nucleotide sequence was analyzed for the different groups. Fixation indices Fst were used to determine genetic differentiation and gene flow between different groups. Tajima’s D test and Fu and Li’s D and F statistics were calculated to evaluate the neutral theory of natural selection, first for all the isolates, and then subsequently on different groups.

The basic nucleotide composition (A%, U%, C%, and G%), AU and GC contents, and relative synonymous codon usage (RSCU) were analyzed using MEGA software.

Recombination events on the P1 virus genome were predicted using RDP v.4.96. A full exploratory recombination scan on the aligned P1 virus genome was performed with seven methods (RDP, Chimaera, BootScan, 3Seq, GENECONV, MaxChi, and SiScan). Settings for both BootScan and SiScan were as follows: BootScan window size = 200 and step size = 20. Additional BootScan settings were the following: bootstrap replicates = 100, random number seed = 3, and cutoff percentage = 70. Parameters used in SiScan were a p-value permutation of 1000 and a scan permutation number of 100. The window size for RDP was set at 30. Variable sites per window were 70 for MaxChi and 60 for Chimaera. Recombination events predicted by at least three of the algorithms with a *p*-value (<0.05) were considered to be valid. The genome recombination events of P1 viruses predicated by RDP software were verified using SimPlot software.

The substitution rate or evolutionary rate was estimated by calculating the number of mutations in the nucleotide sequence of the complete genome, divided by the total number of nucleotides and by the number of years [18,19].

## 3. Results

### 3.1. Genome Properties of P1 Virus Strains in This Epidemiological Investigation

One hundred and sixty sera of the two hundred and forty samples tested positive for the P1 virus; meanwhile, forty-three samples tested positive for PCV2. Twelve P1-positive samples were sequenced, and complete genomic sequences were deposited in the GenBank database under the accession numbers OL792747-OL792758. The sequence analyses revealed that all P1 strains in this study were 648 nt in length, except the JS20-9 strain (OL792755), which has 649 nt. The nucleotide identity analysis of the 12 strains revealed that they share 96.1–100% of their nucleotide identity.

The capsid proteins have 114 amino acids for all P1 strains with a genome of 648 nt, except those of JS20-7 (OL792753) and JS20-8 (OL792754), which have 122 amino acids. The JS20-9 with a genome of 649 nt has a capsid protein of 163 amino acids.

### 3.2. Phylogenetic Analysis

A total of 96 P1 strains were obtained, but the number was reduced to 68 after redundancy analysis. The P1 genome size range is 647–649 nucleotides. The neighbor-joining phylogenetic tree created from the complete genomic nucleotide sequences clustered the P1 strains into two phylogroups: A (n = 64) and B (n = 4). Group A is the dominant, most prevalent genotype. Phylogroup A has 648 (or 647) nt, while phylogroup B has 649 nt. Phylogroup A, including a P1 strain with 647 nt (MG708302), has a nucleotide deletion corresponding to position 207 within the *VP1* gene of phylogroup B. The translation of the *VP1* gene mainly produced a protein of 114 (or 122) amino acids in phylogroup A and 163 amino acids in phylogroup B. Phylogroup A has at least eight subgroups (I-VIII), while the majority of strains did not cluster with any subgroup (Figure 1).

P1 strains from group A shared a nucleotide identity of 96.2–99.8% and, relatively speaking, the nucleotide identity between the two P1 strains from cats (MT318823 and MT318824) and other viruses in this group is lower at 96.8–98.5%. P1 strains from group B shared a nucleotide identity of 94.3–99.8%. P1 strains from group A shared a 92.7–96.1% nucleotide identity with phylogroup B strains.

Subgroup I consisted of two strains (JN040278 and MH167401), subgroup II consisted of two strains (JN207912 and MH167400), subgroup III consisted of six strains (OL792754, MH379143, KU323639, KY462783, KY462787, and MT318822), subgroup IV consisted of two strains (MT318823 and MH167404), subgroup V consisted of two strains (MG708304 and MG708309), subgroup VI consisted of two strains (JN207914 and MH167399), subgroup VII consisted of three strains (OL792751, KU243695, and KU356937), and subgroup VIII consisted of five strains (KU356938, KU350634, MF716583, KU356936, and KU356939).

Nucleotide G at position 170, G at 136, C at 77, G at 376, C at 88, C at 58 and 68, A at 308 and G at 627, and A at 292 were unique to subgroup I, II, III, V, VI, VII, and VIII, respectively. Subgroup III included the most P1 strains encoding a capsid protein with 122 aa. The most predominant nucleotide substitutions identified in subgroup III were T77C, and most strains also have A121G. Although the JS20-7 strain (OL792753) also encodes a capsid protein containing 122 amino acids, it is not included in subgroup III because its base has no above-substitution characteristics.

### 3.3. Sequence Variation and Genetic Diversity

Sixty-eight P1 viruses were successfully selected for genomic analysis. DNA sequence polymorphism site analysis revealed 523 invariable sites and 124 variable sites (including 67 singleton variable sites and 57 parsimony informative sites). The sequence alignment analysis of the sequences suggested that they were classified into two groups, A and B. For group A, 98 single nucleotide polymorphisms (SNPs) were identified at 95 polymorphic sites; for group B, we found 30 SNPs at 30 polymorphic sites. The polymorphic sites of group B are mainly located in 1 to 402 nucleotides, while the polymorphic sites of group A are mainly located in 1 to 201 and 303 to 477 nucleotides (Figure 2).

The genetic diversity and tests of neutrality of the P1 virus were analyzed. The average number of nucleotide differences (K) for groups A and B were 5.078 and 15.167, respectively. A and B showed similar values of haplotype (gene) diversity (Hd, 0.994 ± 0.004 for A and 1.000 ± 0.177 for B). The values of nucleotide diversity (π) of P1 viruses were 0.01272, in which group B (0.02337) was greater than those of group A (0.00785).

The dN/dS values of the P1 viruses were less than one for both group A (0.55) and group B (0.30), suggesting that P1 viruses were affected by negative natural selection.

### 3.4. Nucleotide Composition Analysis

Nucleotide composition analysis revealed that all P1 viruses had the highest mean compositional value of G% (30.0), which was followed by T (U)% (28.7), A% (22.8), and C% (18.5). Furthermore, the mean GC and AU compositions were 48.5 and 51.5%, respectively, which indicated that the P1 viruses were all AU rich.

### 3.5. Relative Synonymous Codon Usage (RSCU) Analysis

The RSCU analysis of the complete coding sequences of Cap revealed that all the overrepresented codons (RSCU value > 1.0) ended with UU/CC, whereas all of the underrepresented codons (RSCU value < 0.5) ended with CG. All P1 viruses analyzed in this study shared the overrepresented codons (UUU, CUU, GUU, AUU, UCC, CCC, ACC, and GCC), and all ended with UU/CC. In addition, all P1 viruses analyzed in this study did not use UUA for leucine, GCA for alanine, AGC for serine, or UAG and UGA for stop codon at all. CCC for proline was identified as the highly preferred codon among the P1 viruses.

The highest RSCU value for the codon was UCC for the serine (4.96) amino acid, and the lowest values were UCG for serine (0.01) and CGG for arginine (0.01) (Table 1).

### 3.6. Gene Flow and Genetic Differentiation between Sub-Populations

A level of gene flow (Nm = 0.22) was observed among the P1 populations, which showed that gene mobility among populations is low. The pairwise Wright’s fixation index (Fst) value for capsid protein between groups A and B was 0.754, revealing the high genetic differentiation of the P1 populations.

### 3.7. Neutrality Test among Sub-Populations

A neutrality test performed on all P1 strains indicated a negative Tajima’s D value −2.272, *p* < 0.01. Fu and Li’s neutrality test for all strains were negative and significance D = −3.289, *p* < 0.02 and F = −3.454, *p* < 0.02, suggesting that the P1 viruses were under-purifying selection. Neutrality testing was further performed on different groups. The value of Tajima’s D for group A was negative (−2.3467, *p* < 0.01). Meanwhile, the value was −0.861 for group B, although this was not significant (*p* > 0.1). The values of Fu and Li’s D and F for each group also showed a similar trend of negative values.

### 3.8. Recombination Events within the Genome of the P1 Virus

Out of 68 strains from the study, 5 strains (OL581721, OL581722, OL581723, OL792755, and MT318823) showed evidence of recombination using the RDP4 software supported by a strong *p*-value in at least three of the detection methods (Table 2). Recombination was found in the genomes of all P1 strains belonging to the B group. P1 KJ612072 was predicted as the parent of four recombinant strains, namely, OL581721, OL581722, OL581723, and MT318823. Some P1 strains (OL581721, OL581722, and OL581723) were predicted to have derived nucleotide fragments of 317–623 from the major parent (KJ612072), and the recombination events were statistically supported by MaxChi, SiScan, and 3Seq methods with average *p*-Value of 2.228 × 10^−4^, 2.446 × 10^−12^, and 1.41 × 10^−6^, respectively.

One strain (OL792755) was predicted to have derived nucleotide fragments of 1–62 and 306–649 from the minor parent (MH167404), and the recombination events were statistically supported by GENECONV, BootScan, MaxChi, Chimaera, and 3Seq methods with average *p*-Value of 3.557 × 10^−2^, 1.851 × 10^−2^, 7.184 × 10^−7^, 2.944 × 10^−3^, and 1.426 × 10^−5^, respectively.

Strain MT318823 was predicted to have derived nucleotide fragments of 218–386 from the minor parent (KJ612072), and the recombination events were statistically supported by MaxChi, Chimaera, and 3Seq methods with average *p*-Value of 2.846 × 10^−3^, 2.230 × 10^−3^, and 4.037 × 10^−4^, respectively.

In addition, a similarity plot analysis was carried out with putative parental sequences using SimPlot software to corroborate the results obtained using the RDP4 software. Taking the strain OL581722 as an example, the similarity plot indicated that the recombinant presented a nucleotide similarity with a P1 strain (KJ612072) between nucleotides 380 and 540 (Figure 3).

### 3.9. P1 Evolution

Nearly 20 years of Chinese P1 virus strains were used for evolutionary analysis. Compared to the sequences of the first submissions to GenBank (EF514716 for group A, and OL581722 for group B), ninety-six mutations were observed in the consensus sequences of group A, and 30 mutations were found in group B. Using the sequence EF514716 from 2005 as the reference, we estimated that the nucleotide substitution rate of group A of P1 viruses varied between 9.65 × 10^−5^ and 7.72 × 10^−4^ subs/site/year. There are few strains in the group B of P1 viruses, and only four strains have been found at present. The evolutionary rate calculated for group B of P1 viruses was higher (1.18 × 10^−4^–2.79 × 10^−3^ mut/site/year) than that of group A. The mean evolutionary rate of P1 viruses was 3.64 × 10^−4^ subs/site/year (Table 3).

## 4. Discussion

The P1 virus has been known for nearly 20 years, but its genetic diversity and evolution in China are less well understood. This work provides an analysis of the diversity of the P1 virus and its evolution as a novel pathogen. In this study, 240 samples collected between 2020 and 2021 were used for investigating P1 and PCV2 infection, and the results showed that the infection rate of the P1 virus is much higher than that of PCV2. The positive rate for P1 was approximately 19% in 2010 [5], increasing markedly to 66.7% in 2021.

A phylogenetic tree based on complete genome sequences of P1 viruses obtained in this study and those retrieved from the GenBank database separated the strains into two main phylogroups: A and B. As the predominant genotype, subgroup A includes at least eight different sub-clusters compared to phylogroup B, which has only one sub-cluster. Out of the twelve strains obtained in this study, eleven belong to phylogroup A, and one belongs to phylogroup B. The comparison of nucleotide diversity from our study indicates a relatively high nucleotide diversity among P1 strains from China. The limited gene flow between groups A and B of P1 viruses in this study revealed that the gene exchange between populations is insufficient, and there are great differences among populations. The phylogroup A in Chinese pigs has been the only pathogenicity-related genotype previously studied [1,2,3]. The phylogroup B has been sporadically present in China, and further studies are needed to compare the virulence using different strains of A and B.

Recombination events were common in populations of phylogroup B compared with those of phylogroup A, indicating that active intragenic recombination is occurring in group B. Reference strain KJ612072 was the parent for most recombinant strains.

Selection pressure analysis showed that PCV1 and PCV2 were under positive selection pressure [20,21], while PCV3 was under negative pressure [22]. The dN/dS values for P1 viruses were negative, implying that negative selection might act in the gene. This is also consistent with the Tajima’s D value observed in our study. The significant negative values of neutrality tests (Tajima’s D and Fu’s Fs) indicated a pattern of recent population expansion in the P1 populations despite the negative but non-significant Tajima’s D value in population B.

Codon usage bias analysis plays an important role in studying virus evolution, and the preference for specific codon usages is a complex evolutionary process associated with several factors, such as compositional constraints, gene function, mutation pressure, natural selection, genome composition, differential mutational pressure, the subgenotypes and aromaticity, and natural selection [23,24,25,26]. As in PCV2 [27] and PCV3 [28], the dinucleotides (UU, CC, or CG) influence the codon usage bias of the P1 virus.

Overall, the P1 populations showed high haplotype diversity and high nucleotide diversity; this may be due to the overexpansion in the population size. The Fst value revealed that group A was genetically different from group B. In contrast, the Fst values among the other eight populations of group A were low, which indicated that they might share the same ancestors. Negative selection and recombination may be the main factors contributing to genetic diversity among P1 strains.

Using EF514716 and OL581722 as the initial reference sequences, we have obtained substitution rates for group A of P1 viruses that are estimated to be between 9.65 × 10^−5^ and 7.72 × 10^−4^ subs/site/year. As with other ssDNA viruses, PCV2 has the highest evolutionary rates, and Firth et al. estimated the rate to be 1.2 × 10^−3^ subs/site/year [29], and the evolutionary rate of PCV2 obtained from a subsequent study (between 3.12 × 10^−3^ and 6.57 × 10^−3^) is similar [20]. In 2019, it was reported that the evolutionary rate of PCV3 was 10^−5^–10^−6^ subs/site/year [22], or between 1.53 × 10^−4^ and 3.40 × 10^−4^ [30]. Overall, the evolution rates of P1 strains were between those of PCV2 and PCV3. The evolution rate of P1 is within the range observed in other parvoviruses [31], geminiviruses such as the maize streak virus [32] and tomato yellow leaf curl virus [33], and nanoviruses such as the faba bean necrotic stunt virus [34]. These substitution rates are within the range of RNA viruses, whose evolution rates vary between 10^−5^ and 10^−2^ subs/site/year [35,36,37].

## 5. Conclusions

Although its genome is relatively small, the P1 virus shows genetic polymorphisms. Two different genotypes, A and B, were detected in the P1 virus, with a higher prevalence of the A type. Negative selection and recombination events were predicted in the P1 virus, which may be the main reasons for the genetic diversity of the P1 population.

## Figures and Tables

**Figure 1 viruses-14-00696-f001:**
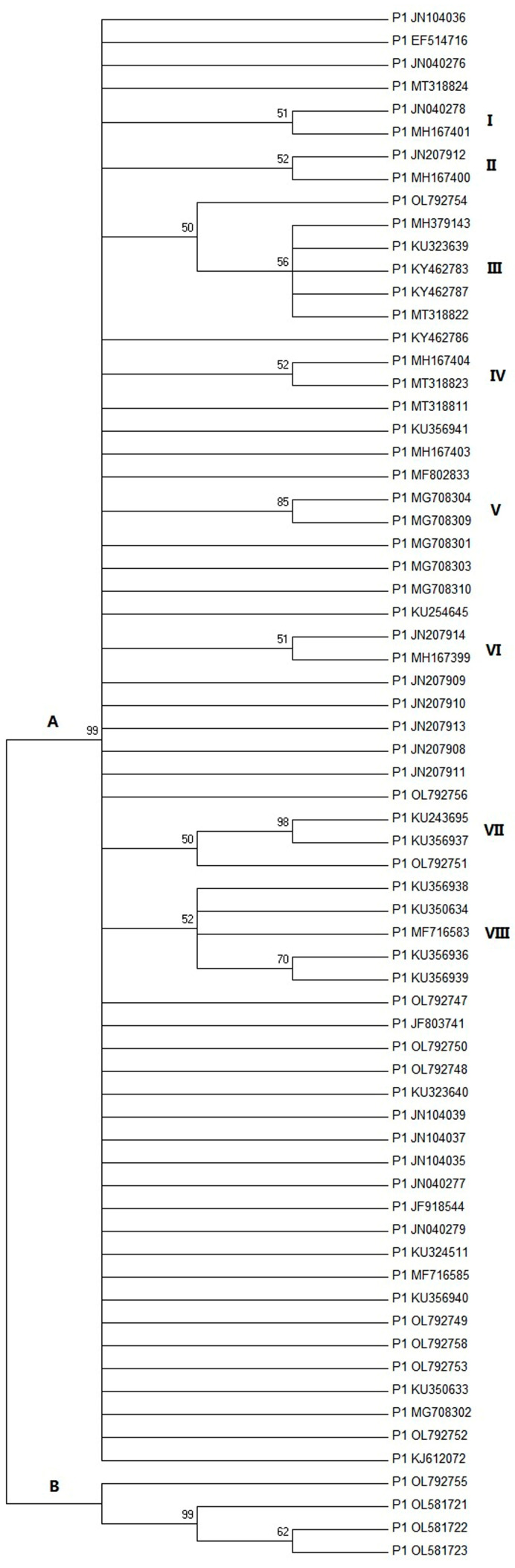
Phylogenetic analysis of 57 complete P1 genomes detected in this study, together with 310 complete P1 genomes deposited in GenBank. The tree was constructed using the neighbor-joining method. The bootstrap consensus tree was calculated from 1000 replicates. Branch support values lower than 50% were not included. P1 strains are represented by their accession number at GenBank.

**Figure 2 viruses-14-00696-f002:**
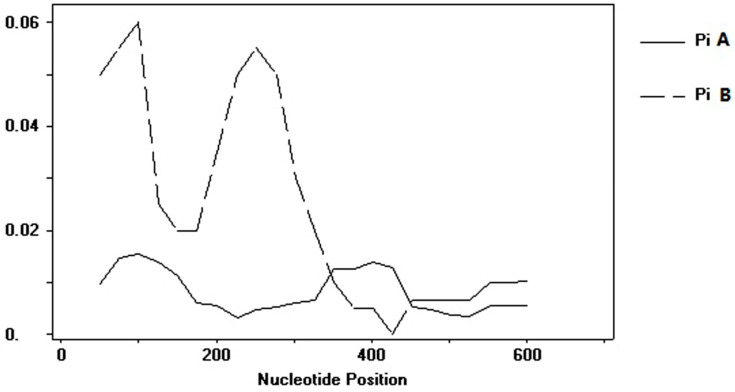
Nucleotide diversity in P1 virus genome. A window size of 100 bp and a step size of 25 bp were used. Group A is presented by broken lines, and B by straight lines.

**Figure 3 viruses-14-00696-f003:**
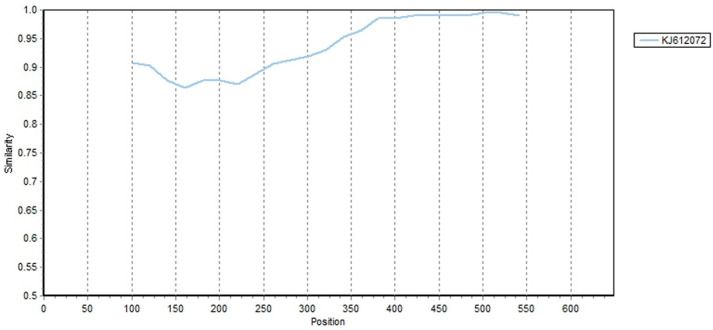
Similarity plot analysis was conducted via Simplot v3.5.1 using a sliding window of 200 nucleotides moving in steps of 20 nucleotides. The genome of strain OL581722 serves as a query sequence. The recombination event is suggested in KJ612072.

**Table 1 viruses-14-00696-t001:** Over-represented codons (RSCU value > 1.0) and under-represented codons (RSCU value < 0.5) in Cap gene of P1 strains.

AA	Codon	RSCU	AA	Codon	RSCU	AA	Codon	RSCU	AA	Codon	RSCU
F	UUU	1.03	S	UCU	0.92	Y	UAU	0.48	C	UGU	0.23
	UUC	0.97		UCC	4.96		UAC	1.52		UGC	1.77
L	UUA	0		UCA	0	-	UAA	3	-	UGA	0
	UUG	0.02		UCG	0.01		UAG	0	W	UGG	1
	CUU	2.78	P	CCU	0.34	H	CAU	0.96	R	CGU	0.83
	CUC	0.37		CCC	2.35		CAC	1.04		CGC	0.87
	CUA	1.43		CCA	0.98	Q	CAA	1.19		CGA	0.02
	CUG	1.4		CCG	0.33		CAG	0.81		CGG	0.01
I	AUU	1.09	T	ACU	0.89	N	AAU	1.01	S	AGU	0.11
	AUC	1.11		ACC	1.84		AAC	0.99		AGC	0
	AUA	0.8		ACA	0.86	K	AAA	1.02	R	AGA	2.6
M	AUG	1		ACG	0.41		AAG	0.98		AGG	1.67
V	GUU	2.41	A	GCU	1.37	D	GAU	0.98	G	GGU	0.61
	GUC	0.02		GCC	2.57		GAC	1.02		GGC	0.75
	GUA	1.1		GCA	0	E	GAA	1.98		GGA	1.23
	GUG	0.46		GCG	0.07		GAG	0.02		GGG	1.41

**Table 2 viruses-14-00696-t002:** Recombination event in P1 strains.

Recomb	Major Parent	Minor Parent	Detection Methods						
			R	G	B	M	C	S	T
OL581721-OL581722	KJ612072	Unknown	-	-	-	+	-	+	+
OL792755	Unknown	MH167404	-	+	+	+	+	-	+
MT318823	Unknown	KJ612072	-	-	-	+	+	-	+

Note: R: RDP; G: GENECONV; B: BootScan; M: MaxChi; C: Chimaera; S: SiScan; T: 3Seq.

**Table 3 viruses-14-00696-t003:** Estimation of nucleotide substitution rates of P1 strains that were obtained and the changes in the consensus sequences with respect to the 2005 submissions’ sequences.

Years	Mutations	No. of Nucleotide Sequences	Nucleotide Substitution Rate (Subs/Site/Year)
2009	4	2	7.72 × 10^−4^
2010	9	7	3.97 × 10^−4^
2011	11	10	2.83 × 10^−4^
2013	2	4	9.65 × 10^−5^
2014	53	6	1.51 × 10^−3^
2015	20	15	2.06 × 10^−4^
2016	7	9	1.09 × 10^−4^
2017	43	18	3.07 × 10^−4^
2018	19	6	3.76 × 10^−4^
2019	17	4	4.68 × 10^−4^
2020	31	10	3.19 × 10^−4^
2021	7	3	2.25 × 10^−4^

Different mutations identified at one position in different sequences were counted only once.

## Data Availability

The data presented in this study are available in the Appendix A here.

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
