# Peer review of "Nearly 20 Years of Genetic Diversity and Evolution of Porcine Circovirus-like Virus P1 from China"

_viruses, 2022, doi:10.3390/v14040696_

Round 1

Reviewer 1 Report

Title: Nearly 20 years of genetic diversity and evolution of porcine circovirus-like virus P1 from China

Authors have investigated porcine circovirus-like virus P1 genetic diversity, evolution, and gene flow, by performing a bioinformatics analysis based on the available nucleotide sequences of the P1 virus. This is an interesting set of valuable data on a virus that has its studies still on infancy. Besides the in silico approach, authors have explored their own set of samples and provided a comprehensive text on the subject. In my opinion the manuscript will be well received by the scientific community. The manuscript is well written. I have only a couple of questions below and advise accepting with minor revisions

On introduction authors define acronym PCV2-associated disease (PCVD), however this is not used anymore. I advise deleting acronym

On introduction, authors state that P1 virus infects various animals, such as pigs, dogs, cats, rabbits, goats, cattle, and yaks [2,10,11]. Is there indisputable data on infection on all these animals or the virus was just found in animals? If the later I would change the expression “infects various animals” to “was found in various animals”.

Also on this, the study reporting P1 in yaks is among those mentioned (2, 10 and 11)?

Did authors sequence by sanger? Bidirectionally? Consensus was built? Please describe for more clarity

Author Response

Thank you very much for your comments, and those comments are all valuable and very helpful for revising and improving our paper. Our alterations are as follows:

On introduction authors define acronym PCV2-associated disease (PCVD), however this is not used anymore. I advise deleting acronym

 “(PCVD)” has been deleted as suggested. (line 35)

On introduction, authors state that P1 virus infects various animals, such as pigs, dogs, cats, rabbits, goats, cattle, and yaks [2,10,11]. Is there indisputable data on infection on all these animals or the virus was just found in animals? If the later I would change the expression “infects various animals” to “was found in various animals”.

 The sentence has been changed as suggested. (line 38)

Also on this, the study reporting P1 in yaks is among those mentioned (2, 10 and 11)?

At present, only the nucleotide sequence of yak circovirus is recorded in an NCBI database.

Did authors sequence by sanger? Bidirectionally? Consensus was built? Please describe for more clarity

“using standard Sanger sequencing” has been added. (line 81); As for these two questions (Bidirectionally and Consensus), please refer to the relevant contents (lines 81-83).

Reviewer 2 Report

The results of this research have show new insights into the evolution of P1 virus.

Author Response

Thank you for your positive comments.

Reviewer 3 Report

Wen et al. here reported genetic features and evolutionary processes of Chinese porcine circovirus-like P1, which is associated with postweaning multisystemic wasting syndrome in pigs, using a bioinformatics analysis and found that two distinct groups (A and B) were detected, with group A P1 viruses being predominant. The writing is well-done and the results were convinced. These findings will facilitate better understanding the genetic and evolutionary processes of P1 viruses. However, some concerns for the manuscript will be needed.

Abstract section, the description of nucleotide substitutions per site per year for P1 viruses should be included rather than that of group A P1 viruses.

Discussion section, whether difference of pathogenicity between P1 groups A and B existed needs to be further discussed.

Author Response

Thank you very much for your comments, and those comments are all valuable and very helpful for revising and improving our paper. Our alterations are as follows:

Abstract section, the description of nucleotide substitutions per site per year for P1 viruses should be included rather than that of group A P1 viruses.

The sentence has been reconstructed according to the suggestions, (lines 26, 257-258) and we have modified the values in Table 3 accordingly.

Discussion section, whether difference of pathogenicity between P1 groups A and B existed needs to be further discussed.

Due to the late report of group B strains, the pathogenicity of these strains has not been studied, but we have added the sentence of explanation in the Discussion section. (lines 277-280)